# Identification of Genomic Variants of SARS-CoV-2 Using Nanopore Sequencing

**DOI:** 10.3390/medicina58121841

**Published:** 2022-12-15

**Authors:** Ionut Dragos Capraru, Mirabela Romanescu, Flavia Medana Anghel, Cristian Oancea, Catalin Marian, Ioan Ovidiu Sirbu, Aimee Rodica Chis, Paula Diana Ciordas

**Affiliations:** 1Discipline of Epidemiology, “Victor Babes” University of Medicine and Pharmacy, 300041 Timișoara, Romania; 2Doctoral School, “Victor Babes” University of Medicine and Pharmacy, 300041 Timișoara, Romania; 3Public Health Authority Timiș County, 300029 Timișoara, Romania; 4Discipline of Biochemistry, “Victor Babes” University of Medicine and Pharmacy, 300041 Timișoara, Romania; 5Center for Complex Network Science, “Victor Babes” University of Medicine and Pharmacy, 300041 Timişoara, Romania; 6Discipline of Pulmonology, “Victor Babes” University of Medicine and Pharmacy, 300041 Timișoara, Romania

**Keywords:** SARS-CoV-2, sequencing, Nanopore, MinION, Ion Torrent

## Abstract

*Background and Objectives*: SARS-CoV-2 is the first global threat and life-changing event of the twenty-first century. Although efficient treatments and vaccines have been developed, due to the virus’s ability to mutate in key regions of the genome, whole viral genome sequencing is needed for efficient monitoring, evaluation of the spread, and even the adjustment of the molecular diagnostic assays. *Materials and Methods*: In this study, Nanopore and Ion Torrent sequencing technologies were used to detect the main SARS-CoV-2 circulating strains in Timis County, Romania, between February 2021 and May 2022. *Results*: We identified 22 virus lineages belonging to seven clades: 20A, 20I (Alpha, V1), 21B (Kappa), 21I (Delta), 21J (Delta), 21K (Omicron), and 21L (Omicron). *Conclusions*: Results obtained with both methods are comparable, and we confirm the utility of Nanopore sequencing in large-scale epidemiological surveillance due to the lower cost and reduced time for library preparation.

## 1. Introduction

During the last two years, the COVID-19 pandemic has been the main concern worldwide, causing socio-economical losses at an unanticipated scale [1]. The causative pathogen is the severe acute respiratory syndrome coronavirus 2 (SARS-CoV-2) [2]; it is a single stranded, positive-sense, enveloped RNA virus of the *Coronaviridae* family, *Betacoronavirus* genus. SARS-CoV-2 is the seventh identified coronavirus shown to infect humans [3]. Its genome was sequenced and published in January 2020 by Wu and collaborators [4]; it comprises 29,903 nucleotides (Genbank-MN908947, RefSeq acc.nos.-NC_045512) [4]. This reference genome has been used for the development of rapid and sensitive molecular diagnostic methods, of which the gold standard is a quantitative real-time polymerase chain reaction (RT-qPCR) [5].

SARS-CoV-2 has a high mutation rate; its genome shows ~1.1 × 10^−3^ substitutions per site each year, which is roughly equivalent to one substitution every 11 days [6]. Consequently, regular viral genome sequencing is needed to obtain information concerning the origin of an infection, the variants circulating in different regions, and the evolution of the spread [2]. Moreover, since these mutations might occur in key regions of the genome, sequencing is needed for the adjustment of the molecular diagnostic assays (primer redesign) and even vaccine development and readjustment [7]. SARS-CoV-2 sequencing could also help in deciphering the link between the mutations occurring and the clinical aspects of the COVID-19 disease [8]. Of note, sequencing proved that reinfections reported shortly after the first infection were caused by a different variant of the virus [9,10].

Today, over 13 million sequences of SARS-CoV-2 have been deposited on GISAID (https://www.gisaid.org/ accessed on 20 November 2022). SARS-CoV-2 sequencing was performed using different approaches: metagenomics, sequence capture or enrichment, and Polymerase Chain Reaction (PCR) amplification using primer/amplicon pools [7,8]. Metagenomics is more appropriate when identifying new, uncharacterized pathogens, without having any knowledge of their genome [11] and is less laborious than other methods. On the other hand, it is relatively expensive, and shows lower depth and coverage for lower viral loads [12]; whole genome sequencing is possible only for samples with a high viral load [11]. In the case of target enrichment sequencing, the knowledge of the genome sequence is essential, because only a predefined region of interest is targeted. These sequences are enriched by hybridization to biotinylated probes or by PCR. Because enrichment occurs even if there is an imperfect complementarity between the samples and the probes, although considered more robust, hybridization is less sensitive to variations in the genome than PCR amplification. Moreover, DNA hybridization enrichment is rather expensive and prone to coverage bias due to competitive binding of the host [11,12]. In both methods, the number of reads depends on the viral load; however, PCR amplification-based methods are better suited in the case of a low viral load or degraded samples [11,13].

In the PCR-tiling amplicon method, one generates pools of amplicons that cover either entirely or partially the viral genome; this can offer sufficient depth and coverage, it is highly specific, and less expensive [8,11]. The approach is not suited to discovering new pathogens, because it requires the knowledge of the sequence, so that the primers can be specifically designed [12]. The major limitation one should consider in PCR methods concerns the primer efficacy, especially when the appearance of new variants is suspected; the interference with the annealing sequences leads to decreased coverage in specific regions or even incomplete assembly.

Short-read sequencing platforms such as Ion Torrent and Illumina are highly accurate and regarded as the current standard [2]. Although long-read sequencing is considered to show lower accuracy, multiple research groups proved that they can be comparable, in terms of coverage and quality of the final sequence, to the short-read sequencing methods [2,8,14,15].

Of the current sequencing technologies available, Oxford Nanopore Technologies (ONT) has gained popularity due to the affordable price and portability of their devices. Nanopore sequencing does not require complex laboratory infrastructure or well-trained personnel, the library preparation procedure is relatively simple, and the results can even be analyzed in real-time [2]. Furthermore, since the Nanopore MinION does not need an internet connection, it can be used for field experiments [12]. The experience with Ebola [16], Zika [17], Lassa fever [18], Yellow fever [19], Influenza [20], and other outbreaks proved that Nanopore rapid sequencing is well-suited not only to epidemic/pandemic surveillance [2], but also transcriptome mapping, identification of new mutations in the viral genome, and characterization of various types of RNA molecules [8].

The ARTIC network (https://artic.network/) has developed a protocol for Nanopore sequencing and data analysis, which was successfully used during Ebola and Zika outbreaks. Shortly after the discovery of the SARS-CoV-2 virus, the protocol was optimized for the identification and characterization of RNA purified directly from nasopharyngeal and oropharyngeal samples. A first pool of primers (V1 primer scheme) was designed to generate 400 bp overlapping amplicons and completely sequence the viral genome. Since then, due to the mutations in the virus’s genome, the protocol has undergone several modifications in terms of alternative primer schemes and amplicon size, which have led to improvements in library preparation (shorter time and up to 96-sample multiplexing) [8,14,15]. Starting at the end of 2020 and early 2021, the V3 primer pool was mainly used. However, new mutations emerged, some of them deletions or single nucleotide polymorphisms (SNPs) in the primer binding sites, leading to amplicon drop-out and incomplete coverage. Hence, yet another set of primers (ARTIC V4) has been designed to solve these issues. The main changes were in the 72_Right primer in G142D (Delta), 74_Left primer in 241/243del (Beta), and 76_Left primer in the K417N (Beta) or K417T (Gamma) (https://community.artic.network/t/sars-cov-2-version-4-scheme-release/312) [21].

The objective of the present study was the setup of workflow for identification by Nanopore sequencing (using a MinION MK1C device and the ARTIC protocol) of different variants of SARS-CoV-2 virus circulating in Timis County, Romania between August 2021 and May 2022. Furthermore, we have compared the Nanopore sequencing results obtained using the V3 and V4 primer pools, and evaluated their quality as compared to Ion Torrent sequencing on an S5 instrument.

## 2. Materials and Methods

### 2.1. Sample Collection and Processing/Ethics

Nasopharyngeal and oropharyngeal swabs were collected from patients infected with SARS-CoV-2 virus hospitalized at the “Victor Babes” Hospital for Infectious Diseases, Timisoara, Romania. All samples were kept in approximately 3 mL of viral transport medium (Shenzen Dakewe Bio-engineering, Shenzen, China) and delivered within 3 h to the Laboratory of Molecular and Biochemical Diagnostic (LDBM) of “Victor Babes” University of Medicine and Pharmacy (VBUMP). All samples were stored anonymized at −80 °C, until further use. The project obtained the approval of the Ethics Committee of VBUMP, Timisoara, Romania.

### 2.2. RT-qPCR Confirmation

Viral RNA was extracted using the Maxwell^®^ RSC Viral TNA Kit (Promega, Madison, WI, USA) on a Maxwell RSC automated machine (Promega, Madison, WI, USA), according to the manufacturer’s protocol. The viral concentration was quantified using an RNA HS Assay Kit (Thermo Fisher, Waltham, MA, USA) on a Qubit 2.0 instrument (Invitrogen, Waltham, MA, USA). All samples were stored at −80 °C until further use. Probes were tested for the presence of SARS-CoV-2 using a GenomeCov19 Detection Kit (ABM, Richmond, BC, Canada) according to the manufacturer’s instructions. All RT-qPCR runs were performed on a Bio-Rad CFX96 instrument (BioRad, Hercules, CA, USA) using a positive and a negative control, targeting N (nucleocapsid) gene, S (spike) gene, and having actin as an internal control. Samples with a Ct over 32 were discarded.

### 2.3. Library Preparation and Sequencing with MinION Mk1C

We used a shorter, adapted version of the SARS-CoV-2 sequencing protocol (nCoV-2019 sequencing protocol v3 (LoCost) V.3) developed and adapted by the ARTIC Network (https://artic.network/) using the reagent from New England BioLabs (NEB, Ipswich, MA, USA) and Oxford Nanopore Technology (ONT, Oxford, UK). Each sequencing run contained 23 samples and a negative control and was performed using the Ligation Sequencing Kit 109 (SQK-LSK109, ONT, Oxford, UK). For cDNA synthesis, 2 uL of LunaScript RT SuperMix (M3010, NEB, Ipswich, MA, USA) was mixed with 8 uL of viral RNA, incubated 2 min at 25 °C, 10 min at 55 °C, and 1 min at 95 °C for enzyme inactivation, then kept at 4 °C until the next step. Samples with Ct between 12–15 and 15–18 were diluted at 1:100 and 1:10, respectively; samples with Ct values between 18 and 32 were used undiluted.

Next, the overlapping amplicons were generated (~400 bp) by mixing 12.5 uL of Hot Start High Fidelity Master Mix (M0494, NEB, Ipswich, MA, USA), 4 uL of primer pool V3 or V4 (ARTIC nCoV-2019 V3 Panel and ARTIC nCoV-2019 V4 Panel, IDT, Corralville, IA, USA), 6 uL of nuclease free water (NFW, NEB, Ipswich, MA, USA), and 2.5 uL of cDNA. Two separate reactions were performed for each sample, using the two primer pools. The cycling program was the following: initial step 30 s at 98 °C, followed by 15 s at 98 °C, and 5 min at 65 °C, for a cycle number between 25 and 35 and cooling at 4 °C. Samples were then pooled together and diluted in 45 μL of NFW (NEB, Ipswich, MA, USA). For the end preparing, we used NebNext Ultra II End Repair/dA-Tailing module (NEB, Ipswich, MA, USA), as follows: 1.2 μL of Ultra II End Prep Reaction Buffer (NEB, Ipswich, MA, USA), 0.5 μL Ultra II End Prep Enzyme Mix (NEB, Ipswich, MA, USA), 5 μL of NFW (NEB, Ipswich, MA, USA), and 3.3 diluted samples from the previous step. The reaction mixture was incubated for 15 min at 25 °C, 15 min at 65 °C, and cooled at 4 °C. Samples were barcoded using 1.25 μL of EXP-NBD104 (barcodes 1–12, ONT, Oxford, UK) and EXP-NBD114 (barcodes 13–24, ONT, Oxford, UK), 5 ul of Blunt TA Ligase Master Mix (M0367, NEB, Ipswich, MA, USA), 3 μL of NFW (NEB, Ipswich, MA, USA), and 0.75 uL of reaction mixture from the previous step, then incubated as follows: 20 min at 25 °C, 10 min at 65 °C, and cooling for 1 min at 4 °C. Next, 10 μL of all barcoded samples was pooled together and purified using 0.4 μL of AMPure XP Magnetic Beads (Beckman Coulter, Brea, CA, USA). Samples were quantified using a Qubit 2.0 spectrophotometer (Invitrogen, Waltham, MA, USA) and dsDNA HS Assay Kit (Thermo Fisher, Waltham, MA, USA). For adaptor ligation, about 30 ng of the barcoded samples was mixed with 10 μL NEBNext Quick Ligation Reaction Buffer (NebNext Quick Ligation Module, E6056, NEB, Ipswich, MA, USA), 5 μL of adaptor MIX (AMII, ONT, Oxford, UK), and 5 μL of Quick T4 DNA Ligase (NebNext Quick Ligation Module, E6056, NEB, Ipswich, MA, USA). The mixture was incubated at room temperature for 20 min, followed by purification with AMPure XP Magnetic Beads (Beckman Coulter, Brea, CA, USA) 1:1 and another Qubit quantification. About 15 ng of the library was loaded in a final volume of 75 μL on a primed R9.4.1 flow cell (ONT, Oxford, UK) fitted in a MinION Mk1C (ONT, Oxford, UK) instrument.

Basecalling and demultiplexing were performed with the MinKNOW 20.10 (ONT, Oxford, UK) (having strict parameters to ensure that amplicons had barcodes at both ends) software, which is integrated in MinION Mk1C (ONT, Oxford, UK). A further data analysis was carried out on the Epi2me platform developed by Metrichor Ltd. (Oxford, UK), Version 2022.04.26-13521, which uses Artic, Nextclade, Pangolin sub-sections (Artic + Nextclade + Pangolin-v3.3.1). The Artic software investigates the depth of coverage for each barcoded sample and can be used for exploring individual amplicons that might not have been amplified with the 2 primer pools. Nextclade software identifies the genetic variants compared to the reference genome and provides data concerning quality control [22]. Pangolin reports the sample lineage [23]. The SARS-CoV-2 virus from Wuhan was used as the reference genome (MN908947).

### 2.4. Ion Torrent Sequencing

The samples were chosen for sequencing based on the quantity and quality of viral RNA (Ct < 32 and minimum 7 ng/μL), and the number of viral copies for each sample was evaluated using the TaqMan 2019-nCoV Assay Kit v1 (Applied Biosystems, Waltham, MA, USA).

The viral RNA was reverse-transcribed using the SuperScript™ VILO™ cDNA Synthesis Kit (Invitrogen, Waltham, MA, USA), according to the protocol provided by the manufacturer. The targets for sequencing were obtained using the Ion AmpliSeq™ SARS-CoV-2 Panel (Thermo Fisher, Waltham, MA, USA). Library preparation was performed with Ion AmpliSeq™ Library Kit Plus (Thermo Fisher, Waltham, MA, USA), on an Ion Chef instrument (ThermoFisher Scientific, Waltham, MA, USA). The samples were sequenced on an Ion Gene Studio S5 instrument (Thermo Fisher Scientific, Waltham, MA, USA), using an Ion Torrent 540 chip (Thermo Fisher Scientific, Waltham, MA, USA). The reads were mapped and assembled using the Iterative Refinement Meta Assembler (IRMA). The alignment, lineage, and clade identification were performed using Nextclade and SARS-CoV-2 as a reference genome (MN908947).

## 3. Results

### 3.1. Nanopore Sequencing

With two exceptions (with Ct of 31.6 and 31.9), all samples had Ct values between 11.9 and 29. For all rounds of sequencing, the average quality score (12.025) was well above the default threshold 7, which indicates a very good sequencing run. Out of the 103 samples loaded, we obtained 96 sequences; the remaining 7 samples had more than 3000 ambiguous reads and could not be unambiguously interpreted. We performed six rounds of sequencing experiments; 23 of the samples were sequenced twice using ARTIC nCoV-2019 V3 Panel (round 4) and ARTIC nCoV-2019 V4 Panel (round 5).

Except for sequencing run 2 (with an overall bad quality), the total number of reads analyzed varied between 705,499 and 4,465,541 (Table 1) with a maximum of 19.95% unclassified reads for the first round. The average number of reads obtained for all rounds was 1,646,673, of which an average of 390,414 were unclassified. The very high number of reads obtained after the first round of sequencing was the result of the longer sequencing time/run (24 h) compared to the other sequencing rounds (6 h). On average, the highest number of reads/clade, 164,439, was obtained for 20I (Alpha, V1), and the lowest for 21K (Omicron) with 39,870 and 21B (Kappa) with 3447 (Figure 1).

With respect to the average number of mutations identified, the highest was recorded for 21L (Omicron) clade-59, followed by 21K (Omicron) clade with 47 mutations, 21J (Delta) having 34 mutations, 20I (Alpha, V1) 32, 21J (Delta) with 26 and the lowest for 20A (24 mutations), and only 1 for 21B (Kappa).

For most of the samples, the overall genome coverage was above 250× with an average sequence length of 519.17 bp. Unexpectedly, the 20 k–22 k genomic region of the virus had a lower coverage, possibly due to sequence alteration in the primers’ binding sites. In Figure 2, there is an example of the coverage obtained for sample barcode 5 from sequencing round 6, for primer pools A and B.

We identified a total of 22 lineages belonging to 7 clades: 20A, 20I (Alpha, V1), 21B (Kappa), 21I (Delta), 21J (Delta), 21K (Omicron), and 21L (Omicron) (Figure 3). Although not all lineages were identified, the clades were still assigned as follows: the unidentified lineages from the first round belonged to 21J (Delta), 21I (Delta); from the second round to 21J (Delta), 21B (Kappa); from the third round 21J (Delta); and from the fifth round to 21K (Omicron) (Figure 3).

In the first round of sequencing (performed on 10.12.2021), we identified 20I (Alpha, V1), 20A for the samples collected in February 2021, and Delta for the samples collected in August—September 2021. Except for only two (Delta) samples, Nanopore results (clades) were confirmed by Ion Torrent sequencing on an Ion S5 instrument (Table 2).

In the second round of sequencing, performed on 21.01.2022, we used samples from October 2021, and the variants most frequently identified were 21J (Delta) and 21I (Delta); two of the samples were classified as 20I (Alpha, V1). For four samples belonging to 21J (Delta), 21B (Kappa) clades, the lineages could not be identified.

In the third round (performed in 10.03.2022) of sequencing, we used samples from November and December 2021, when Delta (AY.120, AY.121, AY.125, AY.39, AY.4, AY.42, AY.43) was still the main circulating variant. In the January and February 2022 samples, the predominant variant circulating was 21K (Omicron) and 21L (Omicron) (BA.1, BA.1.1, BA.2). These samples were sequenced two times, with the V3 primer set (round 4, 12.03.2022) and V4 primer set (round 5, 14.04.2022). For most of the samples, the same lineages were obtained with both sets of primers; in the case of two samples, the sequencing with the V4 pool of primers offered a refinement, as BA.1.1 was diagnosed instead of BA.1 with V3. Nevertheless, we noticed a better-quality score and higher average sequence length when using the V4 primer set. In the last set of samples sequenced in 12.05.2022 (patients from February until May 2022), only 21K (Omicron) and 21L (Omicron) were identified.

### 3.2. Ion Torrent Sequencing

For Ion Torrent sequencing, the average loading of Ion Sphere Particles (IPS) was 91%, 99.5% of which contained the actual patient library. The percentage of polyclonal IPS was 28%, low quality products 11.3%, and there were 0% adapter dimers, so the final library IPS was around 60%. The mean read length obtained was 190 bp. A very high percentage of the total reads (98.6%) was aligned against the reference genome. Overall, these indicators show a very good quality of the sequencing run.

The results obtained for 22 samples from the first sequencing run performed on Nanopore and Ion Torrent are presented in Table 2. For most of the samples, the results were concordant in terms of clade (90.90%) and lineage (72.72%) identification. However, with the exception of the 20I (Alpha) clade (on average, 33.8 mutations on Ion Torrent vs. 32.4 on Nanopore), the Ion Torrent consistently identified more mutations compared to Nanopore sequencing: on average 27.9 and 39.8 mutations in the case of Ion Torrent vs. 24.1 and 30.9 in the case of Nanopore, for 20A and 21I+J (Delta), respectively. In terms of concordance between the two methods, the average value was 80% (Table 2). We also noticed that 2 out of 3 samples with the same number of mutations on both methods, had the same substitutions. A representative example is shown in Figure 4.

## 4. Discussion

With each wave, SARS-CoV-2 has become more infectious and more virulent, underlying the importance of understanding its mutation diversity and rate, which are a major source of vaccine evasion [24,25] and treatment resistance [26]. Whole genome sequencing offers important insights on the evolution of the virus [27]; most of these methods are expensive, laborious, and time-consuming, which represent major disadvantages in the case of an epidemic or pandemic. The ideal sequencing method would need to be rapid, affordable, and easy to perform [2,24]; Nanopore sequencing fulfills all these requirements.

Nanopore sequencing is clearly less time-consuming compared to other platforms: ONT sequencing takes less than 20 h compared to Illumina and Ion Torrent (36 h) [28]. Library preparation for ONT sequencing is very easy and requires far less time than other methods. The workflows (using a modified ARTIC protocol) require 8 h or even 5 h using the Rapid Barcoding Kit [14,29]. In our hands, the entire protocol took approximately 10 h for libraries preparation, and 6 h to run the sequencing. In comparison, samples sequenced on Ion Torrent took around 18–20 h only for library preparation, 10 h run on Ion Chef, and another 4 h run time on the S5 instrument.

In terms of costs, Illumina and Ion Torrent sequencing are generally associated with higher costs and more laborious library preparation [15,28]. By using ONT, costs can be reduced significantly [29], and this is a very important aspect in large-scale sequencing. According to Tyson et al. when using the ARTIC LoCost protocol, the material costs for 24 barcodes on MinION can be reduced to GBP 24.91 pounds/sample (even down to GBP 16.85 for LoCost with one wash); when using 96 barcodes, the method becomes even more affordable, GBP 10.49/sample (GBP 8.47 for LoCost with one wash) [30]. Generally, Nanopore sequencing is considered less appropriate due to its lower read-level sequencing accuracy than the short-read gold standard sequencing platforms, Illumina and Ion Torrent [31]. Given the high mutation rate of the SARS-CoV-2 virus, sequencing accuracy is essential. However, Bull et al. proved that regardless of the high error rates, the sensitivity and specificity of ONT methods can be comparable with short-read sequencing (both >99% comparing to Illumina). Single nucleotide variants (SNVs) were detected with high accuracy, and with very high concordance between ONT and Illumina: 99.66% using the *Nanopolish* pipeline, and 98.83% using the *Medaka* pipeline. However, when it came to detecting small indels, ONT performed rather poorly [2].

The potential of Nanopore sequencing was proven by its large-scale use in several outbreaks such as Ebola and Zika [16,17]. Still, despite the advanced Nanopolish pipeline, which significantly reduces the error rate, some groups obtained a lower genome quality with GridION than Illumina MiSeq, possibly due to the diversity of the variants analyzed [28]. Except for the second round, we obtained overall good quality scores. Furthermore, clade (with two exceptions) and lineage (with seven exceptions) analyses performed on MinION were confirmed and are comparable with the Ion Torrent method. However, there were more mutations detected with Ion Torrent, and the concordance between the mutations identified by the two methods was 80%. In the case of ONT sequencing, we had only two samples with unidentified lineages, but known clades.

The ARTIC protocol has been widely adopted, even in laboratories without extensive next generation sequencing (NGS) experience, thanks to its simplicity, low cost, and very good sensitivity [30]. However, due to the frequent SARS-CoV-2 mutations, protocol adjustments, especially the primer sequences, are needed to improve its performance. The V3 primer scheme was the best choice until the Delta variant appeared, which prompted yet another upgrade, to the V4 primer pool [32]. When comparing the performances of V3 and V4 primer pools on a set of 23 samples, we noticed an overall higher quality score and average sequence length when using the V4 primer set. Improvements in the quality, depth, and resolution of the SNPs related to the usage of V4 primers were also reported by other groups. Of note, G142D amino acid substitution might be underrepresented in Delta variants deposited early, since it was not identified with the V3 primer set [21]. Moreover, Lambisia et al. reported that genome recovery was increased in Alpha, Beta, Delta, and Eta variants when the V4 primer set concentration was augmented [32].

Of note, it has been shown that the wastewater surveillance of SARS-CoV-2 by Nanopore sequencing could be a helpful tool as a forewarning system in the case of a new emerging variant [33,34], underlying that continuous, rapid, sensitive, genomic surveillance (and data sharing) is essential in outbreak monitoring [35,36,37].

## 5. Conclusions

In conclusion, we showed that Nanopore sequencing is a method that is easy to introduce in practice, even by laboratories with limited experience in NGS. It offers a reliable alternative for the rapid, efficient monitoring of COVID-19 outbreaks, at a level of sensitivity comparable to Ion Torrent, but with major advantages such as the ease of library preparation, real-time data analysis, affordable costs, and high accuracy of the results.

## Figures and Tables

**Figure 1 medicina-58-01841-f001:**
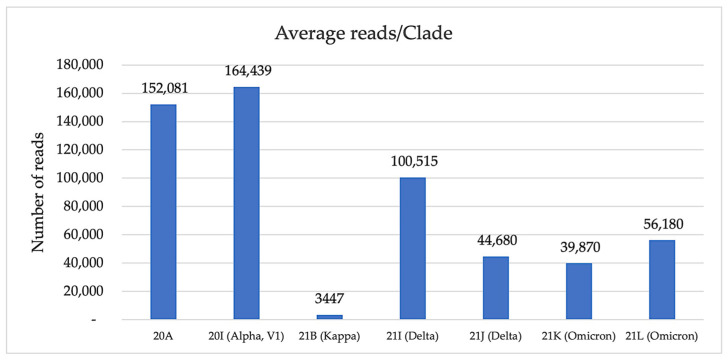
Average read count per each clade.

**Figure 2 medicina-58-01841-f002:**
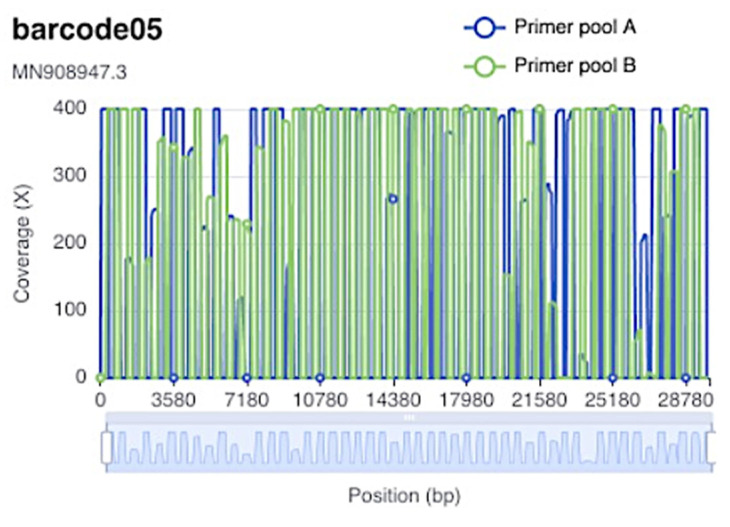
Example of coverage for barcode 5 from round 6.

**Figure 3 medicina-58-01841-f003:**
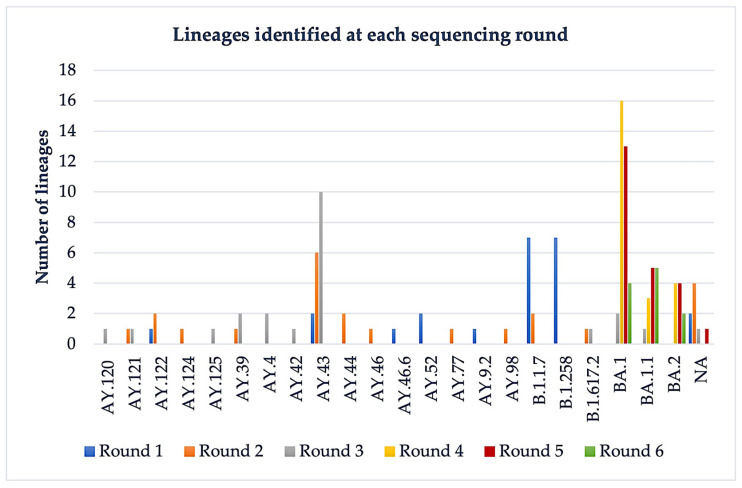
Lineage distribution in the six rounds of Nanopore sequencing.

**Figure 4 medicina-58-01841-f004:**
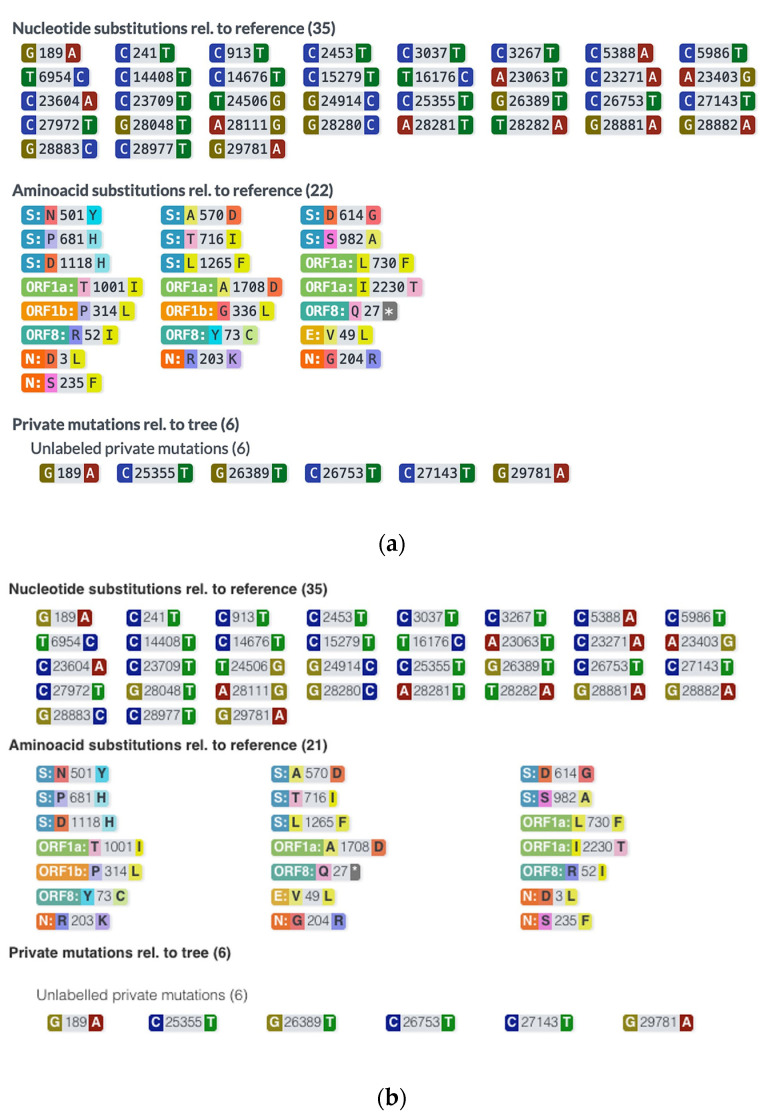
Mutations identified in sample 1 using Ion Torrent (**a**) and Nanopore (**b**).

**Table 1 medicina-58-01841-t001:** Overview of the nanopore indicators obtained.

	Round 1	Round 2	Round 3	Round 4	Round 5	Round 6
Reads Analyzed	4,465,541	1,765,872	998,468	1,095,194	705,499	849,462
Unclassified Reads	890,927	1,214,457	62,643	74,064	52,421	47,970
Total Yield	2.3 Gbases	920.8 Mbases	513.9 Mbases	533.9 Mbases	393.9 Mbases	434.4 Mbases
Average Quality Score	14.1	11.84	11.73	11.32	11.6	11.56
Average Sequence Length (Bp)	524	521	514	487	558	511

**Table 2 medicina-58-01841-t002:** Comparison of the results generated by Ion Torrent and Nanopore sequencing.

Ion Torrent			Nanopore			No. of Common Mutations	Concordance (%)
Clade	Lineage	Total No. of Mutations	Clade	Lineage	Total No. of Mutations		
20I (Alpha, V1)	B.1.1.7	35	20I (Alpha, V1)	B.1.1.7	35	35	100.0
20A	B.1.258.3	32	20A	B.1.258	28	27	84.4
20A	B.1.258	25	20A	B.1.258	20	19	76.0
20A	B.1.258	30	20A	B.1.258	22	21	70.0
21I (Delta)	B.1.617.2	38	21I (Delta)	AY.52	35	33	86.8
20I (Alpha, V1)	B.1.1.7	36	20I (Alpha, V1)	B.1.1.7	33	32	88.9
20A	B.1.258	29	20A	B.1.258	29	28	96.6
20I (Alpha, V1)	B.1.1.7	30	20I (Alpha, V1)	B.1.1.7	30	30	100.0
20I (Alpha, V1)	B.1.1.7	32	20I (Alpha, V1)	B.1.1.7	31	31	96.9
20I (Alpha, V1)	B.1.1.7	34	20I (Alpha, V1)	B.1.1.7	33	33	97.1
21J (Delta)	AY.122	48	21J (Delta)	AY.122	41	41	85.4
20I (Alpha, V1)	B.1.1.7	36	20I (Alpha, V1)	B.1.1.7	34	34	94.4
20A	B.1.258	25	20A	B.1.258	21	21	84.0
20A	B.1.258	21	20A	B.1.258	19	19	90.5
21I (Delta)	B.1.617.2	39	21J (Delta)	AY.46.6	36	16	41.0
21J (Delta)	AY.43	37	21J (Delta)	AY.43	34	34	91.9
21J (Delta)	AY.43	38	21J (Delta)	AY.43	32	32	84.2
20A	B.1.258	33	20A	B.1.258	30	30	90.9
21I (Delta)	AY.9.2	33	21I (Delta)	AY.9.2	27	26	78.8
21J (Delta)	AY.125	45	21J (Delta)	None	21	21	46.7
21J (Delta)	AY.46.6	40	21I (Delta)	AY.52	34	17	42.5
21I (Delta)	B.1.617.2	40	21I (Delta)	None	12	12	30.0

## Data Availability

Not applicable.

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
