# Peer review of "Identification of Genomic Variants of SARS-CoV-2 Using Nanopore Sequencing"

_medicina, 2022, doi:10.3390/medicina58121841_

Round 1
Reviewer 1 Report
Interesting paper and data. The paper will get more strength by increasing by 5 fold the number of samples.
The authors aimed at compairing nanopore and ion Torrent sequencing for SARS-CoV-2 detection and analysis. The results are interesting and they prove that the "rapid" technology can be useful towards this goal during an epidemic. The paper is well writen and presentation is clear. It is interesting to publish these data as a firts report. In future, the data will get more strength and robustness by incresing the number of samples by 5 fold.
Author Response
We thank the reviewer for his comments and suggestions.
The project funded by Romanian National Council for Higher Education Funding, CNFIS (CNFIS-FDI-2021-0484), and by the Romanian Ministry of Education and Research, UEFISCDI (PN-III-P2-2.1-SOL-2020-0142) and was meant to 1) setup the nanopore seq method as a rapid, comprehensive, and alternative to standard second generation sequencing methods and 2) to scan the SARS-CoV-2 variants in the SW of Romania. However, in 2021, the Ministry of Health launched a (centralized) sequencing initiative and collected samples from all over the country that were further processed in Bucharest. This is the main reason why the number of samples we were able to analyze and include in this analysis was rather small. Right now, as our lab has fully implemented the Nanopore sequencing procedures, we are ready to intervene and monitor any further outbreak at the request of the health authorities.
Reviewer 2 Report
This article basically discussed the application of Nanopore sequencing technology into viral surveillance by rapidly detecting the viral mutations and various variants of SARS-CoV-2. This technique was compared to traditional methods and the results were similar. However, several concerns were raised here:
1. The viral mutations detected were regional and limited. How the authors ensure the extension of this study in a larger population-based study? As a high throughput tech, a large size of study object should be considered. If not, would the authors state the rationale?
2. The authors should check through the manuscript and correct the mixup of usages regarding "," and ".". Some numbers stated in the article were apparently ".", but separated by ",". It is confusing and scientifically inaccurate.
3. The references should be improved and more up-to-date articles should be quoted and compared.
4. Figures 1 and 3 should be replotted as an academic drawing is needed.
Author Response
We thank the reviewer for his comments and suggestions.
- The project funded by Romanian National Council for Higher Education Funding, CNFIS (CNFIS-FDI-2021-0484), and by the Romanian Ministry of Education and Research, UEFISCDI (PN-III-P2-2.1-SOL-2020-0142) and was meant to 1) setup the nanopore seq method as a rapid, comprehensive, and alternative to standard second generation sequencing methods and 2) to scan the SARS-CoV-2 variants in the SW of Romania. However, in 2021, the Ministry of Health launched a (centralized) sequencing initiative and collected samples from all over the country that were further processed in Bucharest. This is the main reason why the number of samples we were able to analyze and include in this analysis was rather small. Right now, as our lab has fully implemented the Nanopore sequencing procedures, we are ready to intervene and monitor any further outbreak at the request of the health authorities.
- We apologize for this oversight; we have perused the manuscript and made the necessary changes.
- Thank you for the suggestion; we have updated and extended the reference list with data published on Nanopore sequencing in 2022.
- Thank you for the indication. We added the titles for the axes.